

# Engineering of CHO cells for the production of vertebrate recombinant sialyltransferases

Benoit Houeix and Michael T. Cairns

Glycoscience Group, National Centre for Biomedical Engineering Science, National University of Ireland, Galway, Galway, Ireland

## ABSTRACT

**Background:** Sialyltransferases (SIATs) are a family of enzymes that transfer sialic acid (Sia) to glycan chains on glycoproteins, glycolipids, and oligosaccharides. They play key roles in determining cell–cell and cell-matrix interactions and are important in neuronal development, immune regulation, protein stability and clearance. Most fully characterized SIATs are of mammalian origin and these have been used for in vitro and in vivo modification of glycans. Additional versatility could be achieved by the use of animal SIATs from other species that live in much more variable environments. Our aim was to generate a panel of stable CHO cell lines expressing a range of vertebrate SIATs with different physicochemical and functional properties.

**Methods:** The soluble forms of various animal ST6Gal and ST3Gal enzymes were stably expressed from a Gateway-modified secretion vector in CHO cells. The secreted proteins were IMAC-purified from serum-free media. Functionality of the protein was initially assessed by lectin binding to the host CHO cells. Activity of purified proteins was determined by a number of approaches that included a phosphate-linked sialyltransferase assay, HILIC-HPLC identification of sialyllactose products and enzyme-linked lectin assay (ELLA).

**Results:** A range of sialyltransferase from mammals, birds and fish were stably expressed in CHO Flp-In cells. The stable cell lines expressing ST6Gal1 modify the glycans on the surface of the CHO cells as detected by fluorescently labelled lectin microscopy. The catalytic domains, as isolated by Ni Sepharose from culture media, have enzymatic activities comparable to commercial enzymes. Sialyllactoses were identified by HILIC-HPLC on incubation of the enzymes from lactose or whey permeate. The enzymes also increased SNA-I labelling of asialofetuin when incubated in a plate format.

**Conclusion:** Stable cell lines are available that may provide options for the in vivo sialylation of glycoproteins. Proteins are active and should display a variety of biological and physicochemical properties based on the animal source of the enzyme.

Corresponding author
Michael T. Cairns,
michael.cairns@nuigalway.ie

## INTRODUCTION

Sialyltransferases (SIATs) transfer sialic acid (Sia) to glycan chains on glycoproteins and glycolipids. These sialylated structures are terminal structures on external cell surfaces therefore they are key players in cell–cell, cell-pathogen and cell-matrix interactions, and are important factors in protein stability and clearance (*Bragonzi et al., 2000*; *Choi et al., 2015*; *Olufsen, Brandsdal & Smalas, 2007*), neuronal development (*Galuska et al., 2006*), and immune regulation (*Hennet et al., 1998*; *Varki, 1999*). Some SIATs can also modify lactose to produce the sialyllactoses, 3′ sialyllactose and 6′ sialyllactose, which are present in the new-born milk of many mammals. Human milk contains 12–14 g/L milk oligosaccharides and is particularly rich in sialyllactoses (∼20%). Evidence suggests that sialylated oligosaccharides possess anti-adhesive effects against certain pathogens and that sialyllactoses may increase adhesion of beneficial bacterial strains to the human intestine (*Kavanaugh et al., 2013*; *Lane et al., 2012*; *Mysore et al., 1999*).

Although the sialylation of glycolipids and glycoproteins is universal to most tissues, the sialylation of lactose only occurs in mammary tissue. This process has been the subject of much interest because bovine milk, the source of most infant milk formula (IMF), is much lower in sialyllactoses than human milk (*McJarrow & Van Amelsfort-Schoonbeek, 2004*). Producers of IMF are therefore developing processes to supplement bovine IMF with manufactured sialyllactoses in order to better mirror human milk. Dominant amongst these processes are so called "one-pot" bacterial reaction systems (*Fierfort & Samain, 2008*). These are high yield and cost efficient systems where bacteria metabolize cheap simple sugars to produce sialyllactose. All metabolic pathways within the bacteria that divert from the main pathway are knocked out and novel SIAT genes, not naturally present in many bacterial species, are introduced for the final transfer of Sia to lactose. There are some causes for concern though. First of all, these introduced SIAT genes are bacterial, not mammalian: they carry out the same basic reaction but their protein sequences are only homologous to vertebrate proteins in some short sequence motifs (*Meng et al., 2013*). Bacterial SIATs are present only in select groups of bacteria (CAZy families GT30, GT38, GT42, GT73, and GT80) (*Yamamoto, Takakura & Tsukamoto, 2006*) and many of these strains are highly pathogenic. The natural acceptor structures on bacterial strains are not the same structures as those recognized by vertebrate genes (as their natural role is to add Sia onto bacterial glycoproteins thereby mimicking vertebrate glycans). It is known that some of these SIATs have additional related activities and that they are much more flexible in their choice of acceptor and donor. In situations where, for example, the enzyme is required to add a novel modified sugar to a glycan, this is a distinct advantage; however, in the case of a food, it is essential to maintain high specificity.

It is therefore relevant to investigate the further development of vertebrate SIATs for sialyllactose production and possibly other niche markets. Most commercial SIATs are mammalian in origin, thus the enzymes have adapted to function in an environment of constant temperature, pH, and salinity. As a result, the enzymes are not necessarily

optimal for in vitro reactions. On the other hand, non-mammal species function in much more variable environments and their enzymes will therefore be more variable in their kinetic parameters, substrate specificities, and stabilities (*Siddiqui & Cavicchioli, 2006*). For example, fishes live in conditions of high variability of temperature (0–40 °C) and salinity (freshwater, brackish, or seawater). There is little information on the kinetics and substrate specificities of vertebrate non-mammalian SIATs. In addition, an understanding of the specificity of non-mammalian enzymes to substituted Sia donors would be invaluable in designing new oligosaccharide-based drug analogues. This will become increasingly important as sialosides are targeted in clinical applications (*Bauer & Osborn, 2015*).

Sialyltransferases are subdivided into the ST6Gal, ST6GalNAc, ST3Gal, and ST8Sia families defined by the sugar acceptor (Gal, GalNAc, or Sia) or the specific linkage (α2,3; α2,6 or α2,8) (*Harduin-Lepers et al., 2005*). The SIATs are classical type II membrane proteins residing in the Golgi body. The C-terminal catalytic domain constitutes the bulk of the protein, faces the Golgi lumen and is tethered to the Golgi membrane by a short N-terminal transmembrane sequence. A mostly unstructured stem connects the transmembrane anchor and the catalytic domains. The sequence homology across all the sialyltransferase proteins is strong only in very specific regions (L, S, III, and VS motifs) within the catalytic domain, although within any specific family the homology extends to additional motifs (*Harduin-Lepers, 2010*). The L-motif, associated with donor (CMP-Neu5Ac) binding, is not surprisingly well conserved across SIATs as all use the same donor (*Datta & Paulson, 1995*). Although it was once thought that bacterial SIATs were unrelated to animal SIATS, it is now evident that there is low level amino acid homology at least in the L- and S-motifs, and definite structural homology (*Meng et al., 2013*; *Rao et al., 2009*). The crystal structures of pig ST3Gal I (*Rao et al., 2009*), human ST6Gal I (*Kuhn et al., 2013*), rat ST6Gal I (*Meng et al., 2013*), human ST8Sia III (*Volkers et al., 2015*), human ST6GalNAc II (*Moremen et al., 2018*), and several bacterial SIATs have been recently determined (see review *Audry et al., 2011*). In evolutionary terms ST6Gal enzymes exist in all levels of the animal kingdom back to the echinoderms and hemichordates (*Petit et al., 2018*). In higher animals both ST6GAL1 and ST6GAL2 paralogs exist, however, ST6Gal II has a much more restricted tissue distribution than ST6Gal I which is ubiquitously expressed. The earliest ST3Gal enzymes are found in echinoderms (*Lehmann et al., 2008*) and sponges (*Petit et al., 2015*). There are at least eight ST3Gal paralogs (*Teppa et al., 2016*) with different kinetics and substrate specificities.

The rat, human, and mouse enzymes are probably the best kinetically characterized of the SIATs. However, they are difficult enzymes to assay directly and there are many natural acceptor substrates (oligosaccharides and glycoconjugates) and several potential Sia donors that could be assayed. Sia is a generic term for substituted derivatives of the nine-carbon sugar neuraminic acid. The major Sia donors of biological significance are Neu5Ac (*N*-acetylneuraminic acid), Neu5Gc (*N*-glycolylneuraminic acid) and Kdn (2-keto-3-deoxy-D-glycero-D-galacto-nononic acid) but probably the most clinically relevant is Neu5Gc. Neu5Gc is not synthesized in humans because the enzyme that

**Table 1 Summary of all constructs.**

| Protein | Clone | Species | Accession no. | Catalytic start | SIAT (first 15 residues) |
|---------|-------|---------|---------------|-----------------|--------------------------|
| ST6Gal I | hST6 | Human | NP_003023.1 | Val-63 | VSSSSTQDPHRGRQT |
| ST6Gal I | hST6 A5 | Human | NP_003023.1 | Val-63 | VSSSSTQDPHRGRQT |
| ST6Gal I | hST6 B1 | Human | NP_003023.1 | Val-63 | VSSSSTQDPHRGRQT |
| ST6Gal I | zST6 | Zebrafish | NP_001003853.1 | Val-63 | VKVLRGTGGSKPMYT |
| ST6Gal I | rST6 | Rat | P13721.1 | Val-60 | VFSNSKQDPKEDIPI |
| ST6Gal I | sST6 | Stickleback | CBQ74103.1 | Thr-136 | TLFGGRRRGELSGRV |
| ST6Gal I | fST6 | Fugu | NP_001027933.1 | Thr-136 | TLFGGRRKGELSGRG |
| ST6Gal I | cST6 | Chicken | XP_015132322 | Gln-69 | QMPKALPNNQNKVKG |
| ST6Gal II | hST6Gal2 | Human | NP_115917 | Val-173 | VKKRHRRQRRSHVLE |
| ST3Gal IV | hST3 A1 | Human | AF525084_1 | Ala-47 | AESKASKLFGKLSPL |
| ST3Gal IV | hST3 A2 | Human | NP_001241686 | Ala-47 | AESKASKLFGNYSRD |
| ST3Gal IV | zST3 H6 | Zebrafish | NP_001076498 | Glu-55 | ENLNLNMSRKPELFL |

**Note:**
All fusion proteins start with the Igκ leader sequence followed by the Gateway linkers and the catalytic domain of the sialyltransferase (SIAT).

converts Neu5Ac to Neu5Gc is absent. However, other animals make Neu5Gc therefore ingested meat, xenograph transplants and therapeutic proteins produced by animal cells in culture expose humans to this Sia. The human immune system has been shown to recognize as foreign glycoconjugates containing Neu5Gc.

We have cloned SIAT genes from a variety of vertebrate species and expressed these as secreted proteins (ST6Gal I and ST3Gal IV) in CHO cells. The tagged and purified proteins were assayed for activity in a phosphate-linked assay, for ability to bind appropriate lectins and for production of sialyllactoses.

# MATERIALS AND METHODS

## Materials

All materials were purchased from Sigma-Aldrich Merck (Gillingham, UK) unless otherwise stated. Vectors pCR8/GW/TOPO and pSecTag/FRT/V5-His-TOPO, and the Gateway (GW) vector conversion system were obtained from Invitrogen (Carlsbad, CA, USA). LR Clonase II and the Flp-In-CHO cell line were also from Invitrogen (Carlsbad, CA, USA). FITC-labelled lectins SNA-I and MAA were from EY Laboratories (San Mateo, CA, USA) and biotinylated SNA-I, RCA-I, and MAA lectins were from Vector Laboratories (Peterborough, UK). Oligonucleotide primer synthesis and DNA sequencing was carried out by Eurofins. Proof-reading Phusion High-Fidelity DNA Polymerase (New England Biolabs, Ipswich, MA, USA) was used for PCR amplification of cDNA during construction and standard Taq polymerases (Promega, Madison, WI, USA) for screening. PNGase F was from New England Biolabs (Ipswich, MA, USA).

## Construct design

Protein sequences (Table 1) were aligned in MEGA-X (Fig. S1). Only the soluble, cytoplasmic catalytic domains of the SIATs were cloned because we wanted these proteins to be secreted into the media. In order to make all the cloned proteins as comparable

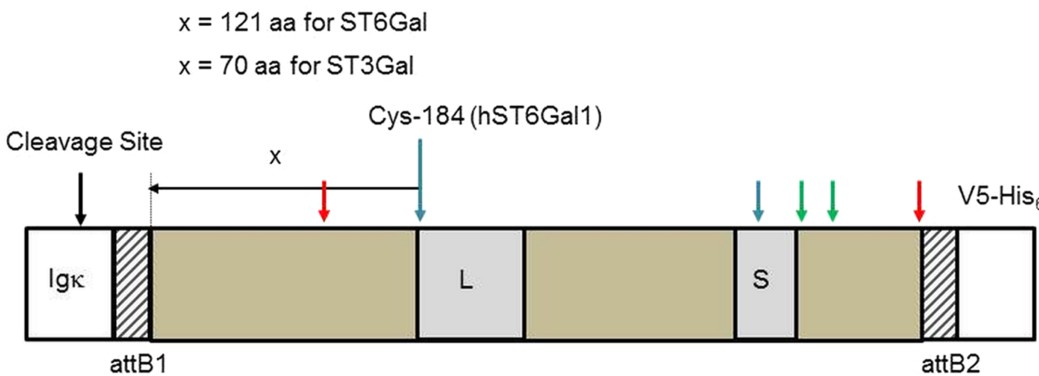

**Figure 1 Sialyltransferase fusion construct detail.** The sialyltransferase constructs (ST6Gal I, ST6Gal II, and ST3Gal IV) all have this general structure. The specifics relate to human ST6Gal I. An Igκ leader sequence is fused in frame with an attB1 sequence (consequence of Gateway recombination), the SIAT sequence, an attB2 sequence and the V5-His₆ C-terminal tag. L and S motifs, common to all SIATs are indicated. Cysteine residues (human ST6Gal I) are paired 142/406 (red arrows), 184/335 (blue), and 353/364 (green). Cys-184 is the reference point chosen to select the start of the SIAT catalytic domain: for ST6Gal I proteins this was 121 amino acids (aa) toward the N-terminal of the protein and for ST3Gal IV proteins this was 70 amino acids (aa) toward the N-terminal of the protein.

as possible we defined the catalytic domains of ST6Gal I and ST6Gal II as starting 121 amino acids N-terminal to a Cys that is totally conserved across all vertebrate SIATs, and which is part of a disulfide bridge that links the L- and S-motifs (Cys-184 in human ST6Gal I NP_003023.1) (Fig. 1). In addition there will be linking amino acids as a consequence of the recombination sequences that are part of the GW cloning system (Table S1). Most proteins have identical sequences through their N-terminal fusions to the SIAT catalytic domain (ending SEFAL). Slight variations in ST6Gal I (zebrafish) fusion sequences ending at KLAL, ST6Gal I (stickleback) ending at AGS and ST3Gal IV (zebrafish) with an additional QKKW sequence are a consequence of an evolving cloning approach (Table S1) and are explained further below. For comparative purposes we have also constructed full-length human and zebrafish ST6Gal I expression vectors. These constructs, based on pcDNA5/FRT/V5-His-TOPO (Invitrogen, Carlsbad, CA, USA), do not support active secretion from CHO cells, however, the catalytic domain of ST6Gal I is detected in cell culture supernatants (*Donadio et al., 2003*; *Monaco et al., 1996*).

Human ST6GAL1 cDNA was sourced from the HepG2 cell line while human ST6GAL2 was sourced from the RPMI cell line. Other STGAL cDNAs were isolated from species include mammals—rat (*Rattus norvegicus*); birds—chicken (*Gallus gallus*); fish—fugu (*Takifugu rubripes*), stickleback (*Gasterosteus aculaeatus*), and zebrafish (*Danio rerio*). Total RNA (2 μg) was isolated from appropriate tissues or cells using the RNeasy Kits (Qiagen, Hilden, Germany) following the manufacturer's recommended protocol (Table S2). Contaminating genomic DNA was removed using an on-column DNase I treatment using an RNase-free DNase Kit (Qiagen, Hilden, Germany). RNA was reverse-transcribed using Superscript VILO (Invitrogen, Carlsbad, CA, USA) as detailed in the manufacturer's protocol.

## Cloning

Primers were designed to each SIAT cDNA ensuring that, after recombination, the gene would be fused in-frame to the N-terminal Igκ secretion signal sequence as well as to the C-terminal V5-His epitope and tag. SIAT fragments were amplified by PCR from cDNA, gel purified and ligated into a GW entry vector (pCR8/GW/TOPO). The GW vector conversion system (Invitrogen, Carlsbad, CA, USA) was used to convert pSecTag/FRT/V5-His-TOPO to a corresponding pSecTag/FRT/GW destination vector. In brief, a GW cassette containing attR recombination sites flanking a *ccd*B gene and a chloramphenicol-resistance gene, was amplified by PCR using specific primers and ligated into pSecTag/FRT/V5-His-TOPO vector. Ligations were transformed into *ccd*B survival competent cells, allowing propagation of the GW destination vector containing the *ccd*B gene. SIAT cDNA fragments were then transferred to this destination vector (pSecTag/FRT/GW) by LR recombination. LR Clonase (Invitrogen, Carlsbad, CA, USA) II was used to catalyze the in vitro recombination between the entry GW clone (containing SIAT flanked by attL sites) and the GW destination vector (containing attR sites). After recombination these sites become attB1 and attB2 sites. Clones were sequenced to confirm in-frame fusion with the N-terminal leader and C-terminal tags. As noted above, there were some variations to this routine procedure leading to some linking sequence differences. Zebrafish st6gal1 was cloned directly into the pSecTag/FRT/V5-His-TOPO, and stickleback ST6GAL1 was recombined into our pSecTag/GW destination vector from a PCR-generated product containing added recombination sites in a related procedure (*Fu et al., 2008*). Either this led to the absence of a GW sequence in the zebrafish ST6Gal I fusion protein or a shorter GW sequence in the stickleback ST6Gal I protein. Zebrafish ST3Gal IV, for unknown reasons, included an additional short sequence which fortunately maintained reading frame.

## Transfection and culture of Flp-In-CHO cells and generation of stable cell lines

Flp-In-CHO cells (Invitrogen, Carlsbad, CA, USA) were grown in Ham F12, 10% Foetal Bovine Serum (FBS), L-glutamine (2 mM) containing 100 μg/mL zeocin. For stable transfections we used a 9:1 (w/w) ratio of pOG44 (1.8 μg) (Invitrogen, Carlsbad, CA, USA) to vector (0.2 μg) and a 1:3 ratio of total DNA (2 μg) to Fugene 6 (6 μL) (Promega, Madison, WI, USA). The plasmid pOG44 expresses the Flp recombinase that recombines the pSecTag derivative plasmid FLP Recombination Target (FRT) sites with the corresponding sites in the Flp-In-CHO cell chromosome. Transfections were carried out in six-well plates (Nunc, Roskilde, Denmark) containing respectively $3 \times 10^5$ cells. Media was removed from adherent, low passage number Flp-In-CHO cells at approximate 60% confluence and replaced with fresh Ham F12, 10% FBS, 2 mM L-glutamine in the absence of zeocin. After 24 h incubation at 37 °C media was replaced with fresh Ham F12, 10% FBS, L-glutamine (2 mM) containing 600 μg/mL hygromycin (optimized previously). Media was changed every 2 days until foci appeared at approximately day 10 (approximately 99% of non-transfected Flp-In-CHO cells were dead at this stage). At 3 weeks, cells were rinsed with phosphate-buffered saline (PBS), treated with

GibcoTM TrypLE (Thermo Fisher Scientific, Waltham, MA, USA) under standard procedures, resuspended with 2 mL Ham F12, 10% FBS, 2 mM L-glutamine, 600 µg/mL hygromycin and transferred to a T25 flask containing 10 mL of media. These cells were subsequently split every 3–4 days to T75 flasks and stocks were frozen (liq. $N_2$) at passage 5. Once stable lines were established, Flp-In-CHO cells were weaned into ProCHO-AT (Lonza, Basel, Switzerland) containing 150 µg/mL hygromycin for several passages over 4–6 weeks. Cells grew strongly in ProCHO-AT but with both adherent and suspension phenotypes. For upscaling, cells were split and grown in T175 triple flasks (500 cm$^2$) in ProCHO-AT to provide 1 L of media for experiments (5 × 200 mL).

## Lectin microscopy

Cells were seeded (3 mL, $1.2 \times 10^6$ cells) in six-well plates containing sterile coverslips and incubated under standard CHO culture conditions to approximately 60% confluence. PBS-washed cells were fixed with 4% paraformaldehyde. Cells were washed three times with a modified TBS buffer, mTBS (20 mM Tris–HCl, pH 7.2, 100 mM NaCl, 1 mM $MgCl_2$, 1 mM $CaCl_2$), blocked with 1% bovine serum albumin (BSA, ≥99%, globulin-free) (Sigma-Aldrich Merck, Gillingham, UK) for 30 min then washed as before with mTBS. Fluorescein isothiocyanate (FITC)-labelled lectins (EY Laboratories, San Mateo, CA, USA) SNA-I and MAA (containing both MAA-1 and MAA-2) were prepared in mTBS (40 µg/mL), added to coverslips (0.5 mL, 20 µg), and incubated for 2 h at 4 °C in the dark in the absence or presence of 1 mL 100 mM lactose. Plates were washed three times with mTBS and then with PBS. The coverslip with the attached cells was mounted on a glass slide with one drop of ProLong Gold antifade mountant containing 4′,6-diamidino-2-phenylindole (DAPI) (Thermo Fisher Scientific, Waltham, MA, USA) and allowed to dry overnight in the dark. Images were acquired using an upright microscope (Olympus BX53) with violet (autofluorescence/DAPI) and green (FITC) filters.

## Sialyltransferase assay

The Sialyltransferase Activity Kit (R&D Systems, Abingdon, UK) was used essentially as described by the manufacturer and as described elsewhere (*Meng et al., 2013*; *Wu et al., 2011*). Our standard assay conditions, used for comparative purposes across a range of sialyltransferase preparations, were 2-(N-morpholino) ethanesulfonic acid (MES) buffer, pH 6.5 (100 mM), $MnCl_2$ (10 mM), CD73 (50 ng), CMP-Neu5Ac (donor substrate), acceptor and 2 µL sialyltransferase in a total volume of 25 µL. All assays were carried out in triplicate. Acceptors were lactose (2.6 mM) or *N*-acetyllactosamine (LacNAc, 2.6 mM). Routinely we used 0.2 mM CMP-Neu5Ac donor for comparative assays and 1 mM CMP-Neu5Ac (with 8 mM lactose) when determining accurate specific activities ($4\times K_m$ concentrations). The sialyltransferase was 100 ng commercial of ST6Gal I (Sino Biologicals (mouse), Wayne, PA, USA) or varying amounts of crude and purified sialyltransferase preparations. Negative control was purified prostate specific antigen (PSA) produced in parallel, originally from the pSecTag/FRT/V5-His-TOPO Vector Kit. Reactions were incubated for 20 or 30 min at 37 °C. For specific

activity determination a range of enzyme amounts were assayed while keeping both donor and acceptor levels at non-limiting levels.

## Purification

Bovine serum albumin present in FBS was a contaminant in early purification protocols therefore Flp-In-CHO cells were weaned onto ProCHO-AT media. Cell-free media (50 mL) in $1\times$ binding buffer (50 mM Tris–HCl, pH 7.5, 500 mM NaCl) and 1 mL Ni Sepharose Excel resin (GE Healthcare, Chicago, IL, USA) were incubated overnight on a roller at 4 °C. The resin was collected after centrifugation (4,000$g$, 15 min, 4 °C) and packed into columns. Columns were washed with 100 column volumes (CV) binding buffer containing 20 mM imidazole (optimized for protein yield and sialyltransferase activity across a 0–30 mM imidazole range). Binding proteins were eluted with two CV 50 mM Tris–HCl, pH 7.5, 500 mM NaCl containing 500 mM imidazole. Buffer exchange with TBS (20 mM Tris–HCl, pH 7.2, 100 mM NaCl) was carried out using Amicon® 3 kDa spin filters (EMD Millipore, Burlington, MA, USA) by centrifugation at 4,000$g$ for 30 min at 4 °C (5 times). The retentate (approximately 500 µL) was concentrated using a 30 kDa spin filter to approximately 50 µL. Protein concentrations were measured spectrophotometrically by NanoDrop and calibrated against BSA protein standards using the bicinchoninic acid (BCA) protein assay (Pierce, Appleton, WI, USA). Proteins were analyzed in reducing conditions on 12% NuPAGE Novex Bis–Tris gels using the NuPAGE MOPS SDS Buffer Kit (Thermo Fisher Scientific, Waltham, MA, USA). The predicted molecular weights of all secreted sialyltransferase fusion proteins are given in Table S3. Gels were Coomassie or silver stained. A wet transfer protocol to Polyvinylidene difluoride (PVDF) membrane (1 h at 24 V) was used (Invitrogen, Carlsbad, CA, USA). Blocking was with 4% dried skimmed milk in TBS-T (TBS + 0.05 % Tween-20) overnight at 4 °C. Detection of His-tagged proteins used either a two-step protocol with mouse anti-His antibody (Novagen, EMD Millipore, Burlington, MA, USA) (1:1,000 dilution) and a HRP-conjugated goat anti-mouse IgG (1:1,000 dilution), or a one-step protocol using mouse anti-His-HRP (1:1000 dilution) (Sigma-Aldrich Merck, Gillingham, UK). BenchMark His-tagged Protein Standard and PageRuler Prestained Protein Ladder (10–180 kDa) were loaded as molecular weight markers (Thermo Fisher Scientific, Waltham, MA, USA). Detection was either with 3,3′-Diaminobenzidine (DAB) or with Enhanced Chemiluminescence (ECL) (Advansta, Menlo Park, CA, USA).
Where applied Ni Sepharose-purified protein was digested with PNGase F (New England Biolabs, Ipswich, MA, USA) for 1 h at 37 °C as recommended by the manufacturer.

## HILIC-HPLC

Glycans were labelled with 2-amino benzamide (2-AB) (*Bigge et al., 1995*) and separated by hydrophilic interaction liquid chromatography (HILIC) using a GlycoSep N-Plus HPLC column (Prozyme, Ballerup, Denmark) on a Waters Alliance 2695 HPLC system with Waters 2475 fluorescence detection. Flow rate was 0.67 mL/min with 20% 50 mM ammonium formate, pH 4.4 (solvent A) and 80% acetonitrile (Romil, Cambridge, UK). Samples were injected in 80% acetonitrile with a linear gradient of 20–58% solvent A

over 48 min. Sialyltransferase reactions (50 µL scale) were carried out much as described above using excess of donor (CMP-Neu5Ac) and acceptor (LacNAc, Lac or whey permeate) at approximately 4× $K_m$ concentrations. Whey permeate (provided by Glanbia plc) was 79.6% lactose by our estimate. Approximately 900 ng of all SIATs (controls and experimental samples) were assayed. Duration (4 h or 16 h) and temperature (37 or 20 °C) of incubations were varied though 16 h incubation became routine. After reaction 450 µL ice-cold HPLC-grade water was added to the reaction, followed by centrifugation (10 min, 20,000g, 4 °C) through Amicon® spin filters. Eluants were concentrated to dryness ready for 2-AB labelling. Resulting glycans were fluorescently labelled with 2-AB by reductive amination (*Bigge et al., 1995*). In brief, an appropriate volume of 0.35M 2-AB in Dimethyl sulfoxide (DMSO) containing 30% acetic acid was dissolved in sodium cyanoborohydride (1M) at 65 °C. A volume of 60 µL of the preparation was used to label dried glycans. Samples were incubated for 2.5 h at 65 °C. A GlycoClean S cartridge (Prozyme, Ballerup, Denmark) was used to separate labelled glycans from unreacted 2-AB following the manufacturer's instructions. Elution was carried out with 3 × 0.5 mL water and 2-AB labelled glycans were evaporated to dryness using a concentrator. In this HILIC system each sugar has a retention time, stated in glucose units (GU) that is unique for that molecule. Calibration of retention with dextran (20 µg, dextran ladder from *Leuconostoc mesenteroides*) (Sigma-Aldrich Merck, Gillingham, UK) was carried out three times over the course of a run to insure accurate GU values. Fifth polynomial equation resolution GraphPad Prism (GraphPad Software, San Diego, CA, USA) was used to convert retention times into GU (*Marino et al., 2011*). Calibration of quantities was determined for a 10-fold dilution (1 µg–0.1 ng) of lactose, 3′ sialyllactose and 6′ sialyllactose using peak height and/or integrated area. Standards were LacNAc (CarboSynth, Compton, UK), Lac (Sigma-Aldrich Merck, Gillingham, UK), 3′ sialyllactose (Sigma-Aldrich Merck, Gillingham, UK), 6′ sialyllactose (Glycom, Esbjerg, Denmark), and Neu5Ac (Sigma-Aldrich Merck, Gillingham, UK).

**Enzyme-linked lectin assay**

Enzyme-linked lectin assay (ELLA) was based on previous protocols (*Couzens et al., 2014*; *McCoy, Varani & Goldstein, 1983*; *Thompson et al., 2011*). Lectin binding is generally quite weak and requires multivalent binding sites acting together to give strong and specific binding. For this assay, we therefore used asialofetuin (ASF) as the acceptor in preference to the disaccharides Lac or LacNAc. Fetuin is a simple glycoprotein (Fig. S2) with a defined structure offering six potential terminal LacNAc sites and is readily available in its asialylated form. ASF (50 µL per well of a 0.5 µg/mL solution in PBS) was added to Nunc™ MaxiSorp polystyrene 96 well ELISA plates (Thermo Fisher Scientific, Carlsbad, CA, USA) and incubated at 4 °C overnight. Wells were blocked for 2 h at room temperature with 150 µL 0.5% (w/v) polyvinyl alcohol. Wells were washed (four times) with 100 µL mTBS-T (mTBS containing 0.05% Tween 20). Sialyltransferase reactions (50 µL) were prepared containing 50 mM MES pH 6.5, 10 mM $MnCl_2$, 0.2 mM

CMP-Neu5Ac, 2.5 units Antarctic phosphatase (New England Biolabs, Ipswich, MA, USA) and the sialyltransferase preparation to be assayed. We used 100, 200, and 400 ng of commercial mouse ST6Gal I as positive control, 400 ng of our test sialyltransferase or a negative control with no sialyltransferase. Reactions was added to wells and incubated at 37 °C for 16 h. Wells were washed (four times) with 100 μL mTBS-T and biotinylated lectins (Vector Laboratories, Peterborough, UK) were added at 2, 5, or 10 μg/mL of SNA-I, RCA-I or MAA, respectively. After a further 1 h incubation at room temperature wells were again washed (four times) with 100 μL mTBS-T. HRP-Streptavidin (R&D Systems, Abingdon, UK) was added at 1/200 dilution and incubated 1 h at room temperature. Wells were washed (four times) with 100 μL mTBS-T. 3,3′,5,5′-tetramethylbenzidine (TMB) substrate (50 μL) was added followed by incubation at room temperature for 15 min with the reaction stopped by adding 50 μL 2M sulfuric acid. Absorbance was measured at 450 nm on a SpectraMaxR® spectrophotometer (Molecular Devices, San Jose, CA, USA).

### Statistics

Student $t$-tests were used to compare SIAT activities between recombinant PSA produced by the control Flp-In-CHO cell line and all other recombinant SIATs purified from stable cell lines. Mann–Whitney tests were carried out to compare the lectin binding of ASF with all recombinant SIATs. Tests were performed with PASW Statistics 22 (IBM SPSS Statistics, Armonk, NY, USA).

## RESULTS

### Constructs

All known SIATs (CAZy-family 29) have an approximate 250 amino acid catalytic domain that is naturally orientated to the lumen of the Golgi. The full length protein has a transmembrane domain that attaches the catalytic domain to the Golgi membrane through a stem structure that is not essential for catalytic activity. Our focus in this study was on expressing the catalytic domains of the SIATs, however, for comparative purposes we also constructed two full-length ST6Gal I expressing vectors (see Materials and Methods). All catalytic domain-expressing constructs were based on pSecTag/FRT/V5-His TOPO with inserts amplified by PCR from cDNA. In order to make all the cloned proteins as comparable as possible we aligned all the protein sequences and maintained a consistent length peptide linker (part of the stem) between the Igκ secretion signal sequence and the known functional limits of the catalytic domain. ST6Gal I and ST6Gal II therefore started 121 amino acids N-terminal to a Cys that is totally conserved across all animal SIATs (Cys-184 in human ST6Gal I). The first amino acid in the human ST6Gal I was Val-63. For ST3Gal IV we chose to define the catalytic domain as starting approximately 70 amino acids N-terminal to this same invariant Cys. All ST6Gal I, ST6Gal II, and ST3Gal IV expressing constructs are summarized in Table 1 and Fig. 1, with additional details in Tables S1–S3. Some minor differences that predated our final routine cloning approach are explained in the Materials and Methods section.
## Sequences

We cloned and developed stable cell lines of three separate human ST6Gal I sequences. One was the wild-type sequence (NP_003023.1). Clone hST6 B1 had an interesting G327S mutation in a highly conserved region in the S-motif: this Gly is highly conserved across ST6Gal I and ST6Gal II proteins in all species (*Harduin-Lepers, 2010*; *Harduin-Lepers et al., 2005*). Clone hST6 B1 subsequently refers to the codon-optimized and synthesized version of this clone. hST6 A5 had two mutations (I295T and K395R) in less highly conserved regions. The zebrafish clone zST6 had both an H111I and a K187R mutation in comparison to the database sequence (NP_001003853.1). Rat had a R105G mutation. All other ST6Gal I protein sequences matched the database sequences (chicken, fugu, and stickleback). Human ST6Gal II (sourced from the RPMI cell line) had an A269T missense mutation. Human ST3Gal IV clones (hST3 A1 and hST3 A2) were different by a six amino acid insertion present in hST3 A1 but were otherwise identical to the database protein sequences. Some of these variant sequences may have been inadvertently introduced in the amplification process (though proof-reading Taq polymerases were used throughout for cloning) or be common variants in laboratory species, but whatever their origins they were useful for the purposes of this study which was to generate a library of enzymes with potentially different physiochemical properties.

Clones were transfected into Flp-In-CHO cells and selected for with hygromycin. Transfected cDNA should recombine at the corresponding FRT sites engineered into the CHO chromosome of these Flp-In cells, therefore no further selection was carried out to identify highly expressing clones and transfected cells were pooled. The transfected cells were investigated initially for lectin binding. CHO cells do not express ST6Gal I therefore there are minimal NeuAc(α2-6)Gal terminal structures on surface glycans (*Jenkins, Parekh & James, 1996*; *Lee, Roth & Paulson, 1989*; *Monaco et al., 1996*). The *Sambucus nigra* SNA-I lectin has specificity for NeuAc(α2-6)Gal/GalNAc therefore this lectin can be used to identify terminal NeuAc(α2-6) structures present in CHO cells as a result of functional expression of the transfected cDNA (*Smith, Song & Cummings, 2010*). The *Maackia amurensis* MAA lectin (undefined mixture of MAA-1 and MAA-2) on the other hand has specificity for NeuAc(α2-3)Gal/GalNAc (*Geisler & Jarvis, 2011*) therefore this lectin can be used to identify terminal NeuAc(α2-3) which is naturally produced in CHO cells by the activity of ST3Gal enzymes. We have used these two lectins to show that NeuAc(α2-6) structures are much more prominent in transfected CHO cells (ST6GAL1-transfected) and that there is little changes in the NeuAc(α2-3) structures (Fig. 2).

## SIAT assays by phosphate linked assay

Sialyltransferase activity was determined indirectly through a phosphate linked assay. LacNAc is considered the best acceptor for human ST6Gal I in this type of assay. Suitable potential acceptors were LacNAc, ASF, or Lac. As production of sialyllactose was the focus of our study, Lac or LacNAc (as disaccharides) were preferred over ASF (a glycoprotein). Samples assayed were predominantly His-purified proteins isolated from

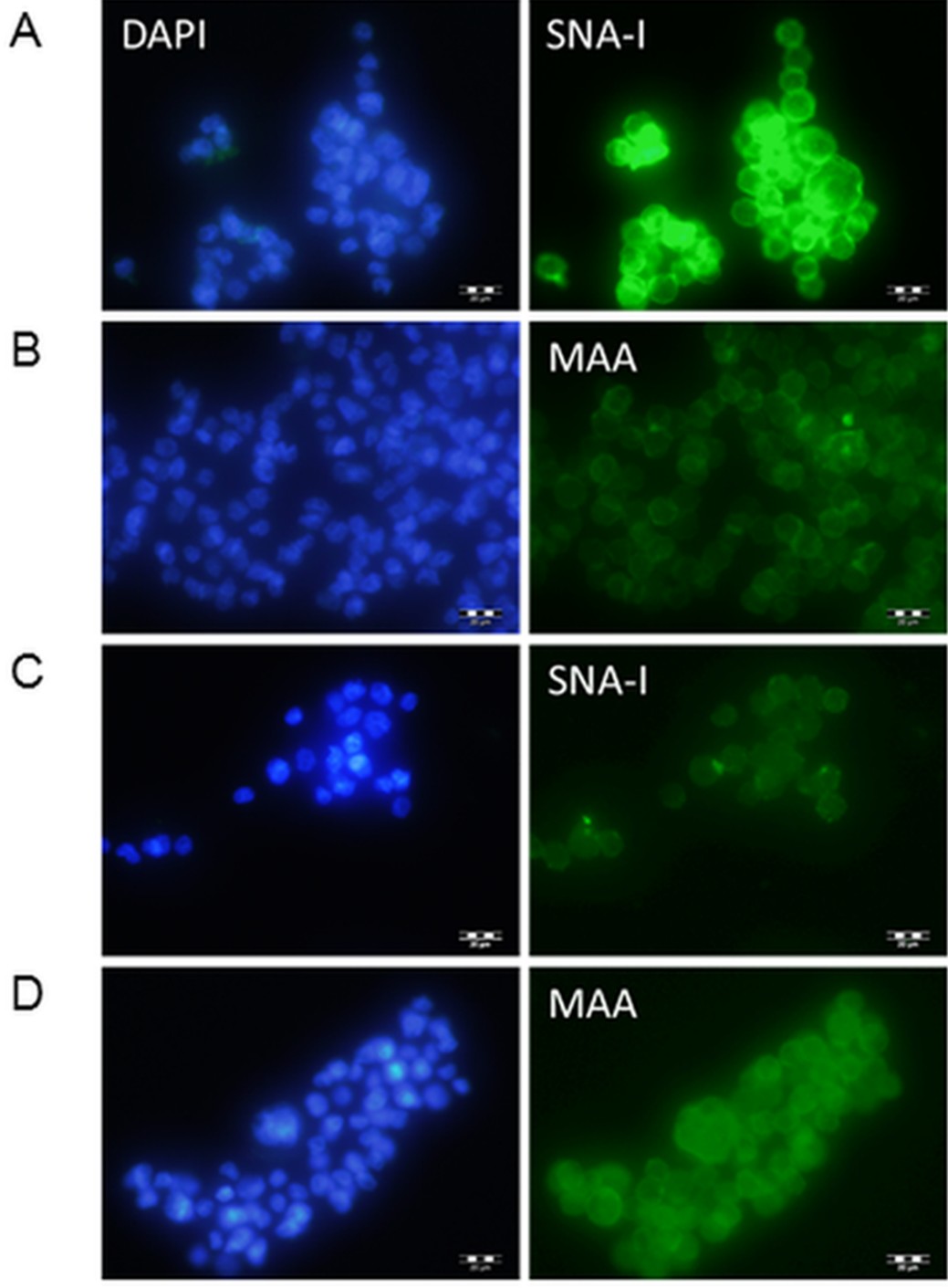

**Figure 2 Lectin binding to CHO cells expressing ST6Gal I.** CHO cells expressing ST6Gal I show strong binding of SNA-I (A) and moderate binding of MAA (B). CHO cells (non-transfected control) showed weak binding of SNA-I (C) and moderate binding of MAA (D). Lectins were FITC-labelled and cells were counterstained with DAPI. Binding of SNA-I was inhibited by the presence of 100 mM lactose (Fig. S3).

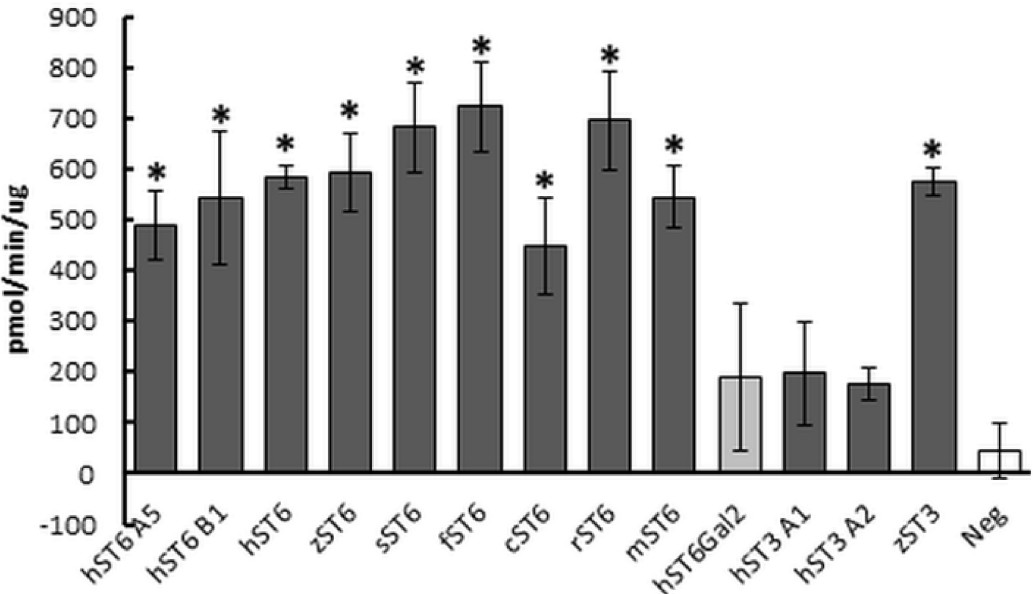

**Figure 3 Sialyltransferase assay on purified enzyme preparations.** Proteins purified on Ni Sepharose Excel (150 ng) were assayed using LacNAc (2.4 mM) as acceptor and CMP-Neu5Ac (0.2 mM) as donor. Reactions were incubated for 30 min at 37 °C. Transfer of Neu5Ac was measured by release of phosphate using a phosphate linked assay with Malachite Green detection. These are the averages of three separate experiments. Human ST6Gal II is shown with lighter shading and the negative control (Neg) (IMAC-purified PSA) has no shading. Mouse ST6Gal I (mST6) is a commercial preparation used as a positive control. See Table 1 for explanation of all clone names. Significant activities following Student $t$-tests between PSA control and SIAT clones are marked with an asterisk ($p < 0.01$).

the media of stably transfected cells, though initially cell lysates were used instead of secreted protein. We determined significant sialyltransferase activity for purified ST6Gal I proteins of all species which were comparable to the commercial control (mouse ST6Gal I, mST6 in Fig. 3). Both mutant human ST6Gal I proteins (hST6 B1 and hST6 A5) were active. Of the ST3Gal IV preparations, only zebrafish ST3Gal IV (zST3) showed significant activity. The two human ST3Gal IV clones (hST3 A1, hST3 A2) and the human ST6Gal II clone showed marginal activity in this assay. The results may be influenced by the choice of acceptor: it is possible that these constructs, showing marginal activity with LacNAc, would show stronger activity with ASF as acceptor. For several enzymes we also measured specific activity across a range of protein amounts. In a direct comparison of activity in cellular lysates human ST6Gal I clones hST6 A5, hST6 B1, and zebrafish clone zST6 gave activities of 0.6 pmol/min/µg, 7.1 pmol/min/µg, and 6.2 pmol/min/µg, respectively. In these samples activity may be influenced by the efficiency of secretion with a poor secretor giving a proportionally higher level of protein in the lysates.

## Protein purification

Proteins were purified as described in Materials and Methods. Because of purification issues around bovine serum albumin present in FBS, cells were weaned onto a serum-free

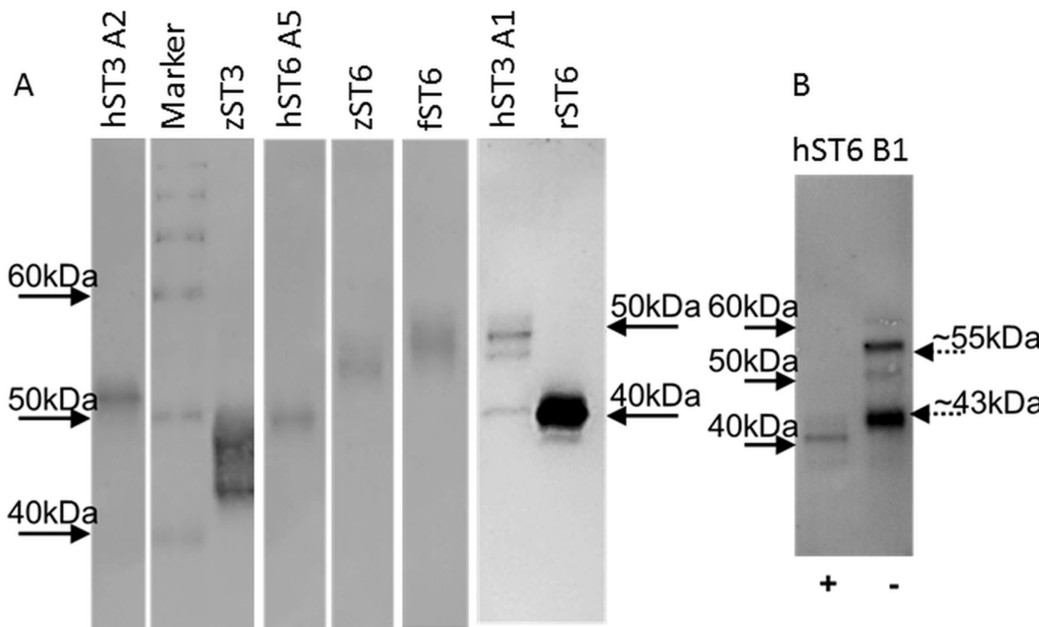

**Figure 4 Western blots of purified proteins.** (A) Various ST6Gal I and ST3Gal IV purified proteins on Western blots detected by mouse anti-His, HRP-goat anti-mouse IgG with DAB staining. (B) PNGase F digested ST6Gal I (+) and undigested (−) control. Loading was ~2 μg purified from ProCHO-AT media separated on 12% NuPAGE Novex Bis-Tris gels in MOPS buffer under reducing conditions. Marker: BenchMark His-tagged protein standard. See Table 1 for explanation of all clone names.

media (ProCHO-AT). A consequence of the switch to ProCHO-AT was that the CHO cells were not fully adherent to culture plates. This was irrespective of whether the Flp-In-CHO cells were transfected or not. Although we did not determine whether the two phenotypes (adherent and non-adherent) differed in their recombinant sialyltransferase expression characteristics, we did show that the phenotype was not fixed—non-adherent cells returned to a mix of adherent and non-adherent cells when reseeded.

Typical protein yields from 200 mL media ranged from 1 to 33 μg across clones. This corresponded to a typical recovery of 2–5% though estimates of protein concentration in crude media were inconsistent. For comparative purposes sialyltransferase assays were carried out with a fixed volume of recombinant enzyme and corrected for protein concentration. Specific activities for select preparations of cell lysates, media or IMAC-purified proteins were determined across a range of protein amounts. Specific activities of 1,697 pmol/min/μg (for commercial mouse ST6Gal I) and 3,147 pmol/min/μg (for human ST6Gal I, hST6 A5, His-purified from a cellular lysate) corresponded to activity measurements for both of approximately 450 pmol/min/μg in the comparative assay (fixed protein) of Fig. 3. Purified proteins were silver-stained on SDS/polyacrylamide gels and were analyzed on Western blots using anti-His antibodies (Fig. 4A). Typically bands on gels were weak and varied in size depending on the species. Glycosylation (heterogeneous) also led to diffuse bands for several proteins and the suggestion of different prominent glycoforms. PNGase-F digestion of purified human ST6Gal I showed a
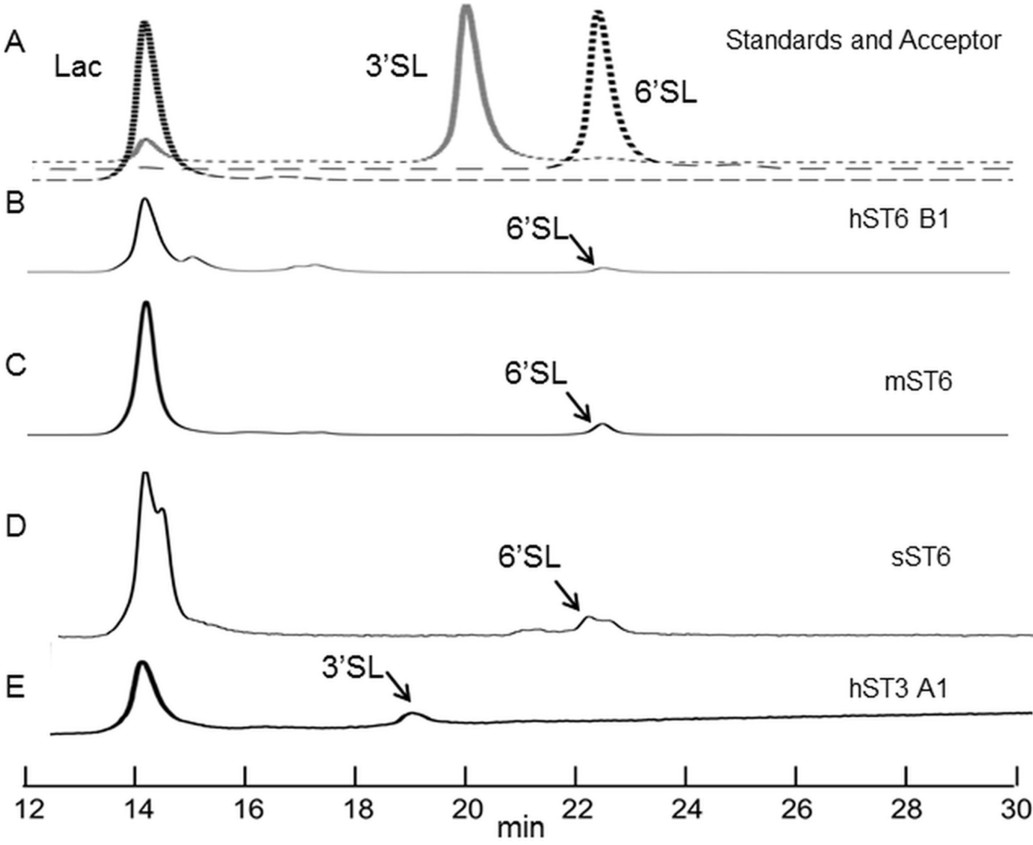

**Figure 5 HILIC chromatographs for typical sialyltransferase incubations.** (A) Overlay of 2-AB labelled whey permeate (~80% lactose), 3′ sialyllactose (3′SL) and 6′ sialyllactose (6′SL) standards. (B)–(E) 2-AB labelled glycans after incubation of whey permeate with various sialyltransferases. Reaction conditions were 16 h at 37 °C (or 20 °C) at 8 mM lactose (in whey permeate) and 2 mM CMP-Neu5Ac. (B) Human ST6Gal I clone B1 (hST6 B1) at 20 °C. (C) Commercial ST6Gal I (mST6) (Sino Biologicals). (D) Stickleback ST6Gal I (sST6). (E) Human ST3Gal IV clone A1 (hST3 A1). Peaks have consistent and diagnostic GU (glucose unit) values calibrated to a dextran standard across runs.

shift downward in molecular mass from 55 kDa (for a prominent glycoform) to 41 kDa (Fig. 4B). There were two prominent glycoforms in this preparation of 55 and 43 kDa, with two weaker glycoforms of 60 and 50 kDa.

## HILIC

Indirect evidence from the phosphate linked assay suggested the presence of sialyltransferase activity. Direct evidence was obtained by incubating the enzyme with lactose and detecting the expected products (3′ sialyllactoses for ST3Gal IV and 6′ sialyllactose for ST6Gal I). The products were labelled with 2-AB and separated by HILIC. Each sugar has a retention time, which when stated in GU calibrated against a dextran ladder, is unique for that molecule. Sialyllactose standards generated distinctive GU values of 2.88 and 3.35 for 3′ and 6′ sialyllactose, respectively.

Reactions were carried out routinely with 8 mM lactose (or whey permeate equivalent) and 2 mM CMP-Neu5Ac which approximate to 4× $K_m$ concentrations (for human

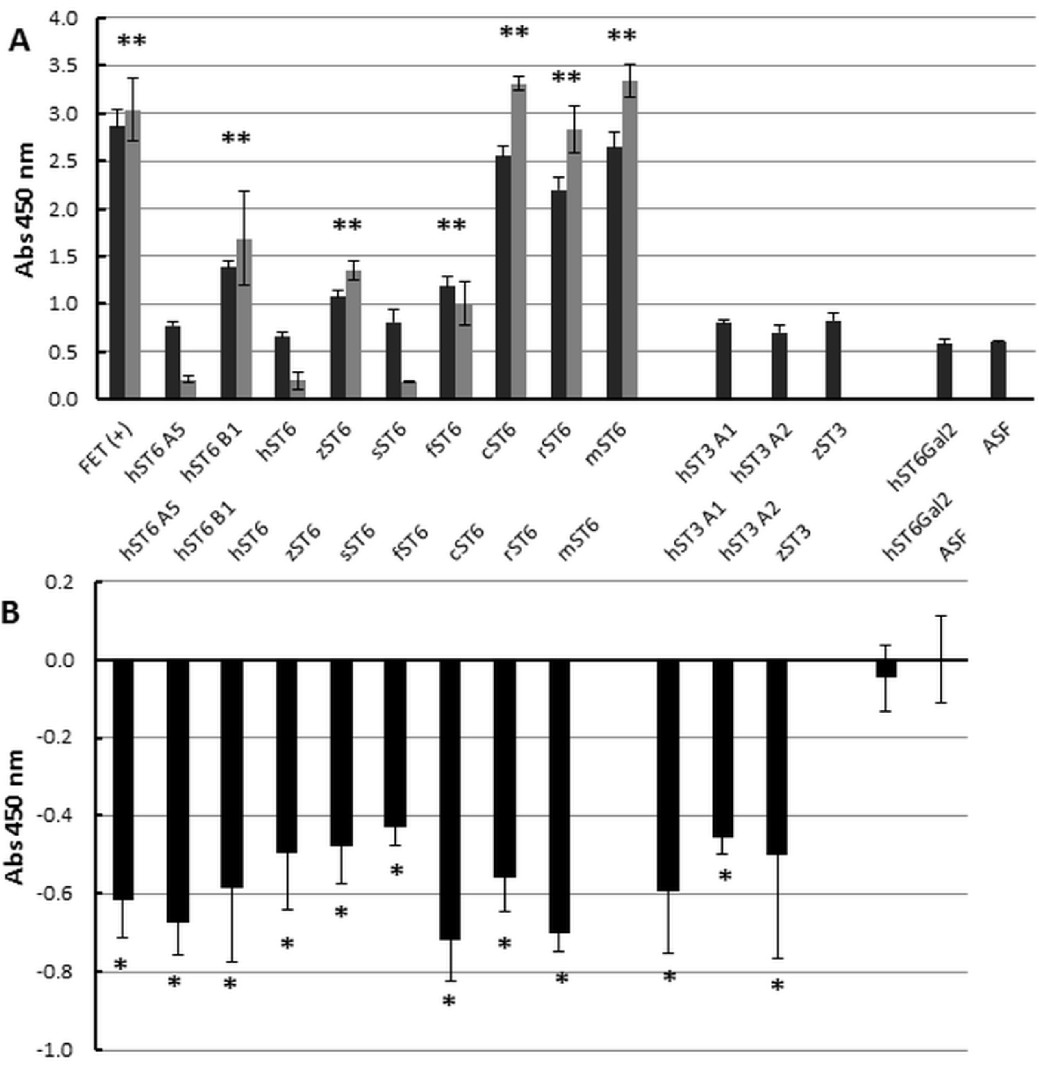

**Figure 6 Enzyme-linked lectin assay.** (A) ASF, reacted with each of the sialyltransferase preparations (37 °C, 16 h), was bound to biotinylated SNA-I lectin. Binding to Neu5Ac on the ASF was detected with HRP-streptavidin. Expt 1 (black bars) included ST6Gal I, ST3Gal IV, and ST6Gal II preparations. Expt 2 (grey bars) included only ST6Gal I preparations. (B) Each sample was analyzed for binding with biotinylated RCA I lectin. All values are relative to ASF. RCA I will bind strongly to Gal residues on ASF but binding will decrease as Gal residues are sialylated with each of the ST6Gal I, ST6Gal II, and ST3Gal IV sialyltransferases. Each sample was assayed in triplicate and error bars show SD. FET, fetuin (positive control); ASF, asialofetuin. Significant binding events following Student *t*-tests are marked with an asterisk ($p < 0.01$).

ST6Gal I), ensuring that both acceptor and donor were in excess. Yields were checked at 1, 4, and 16 h at 37 °C but were subsequently set at 16 h. Both 3′ and 6′ sialyllactoses peaks were evident at the expected positions depending on the purified enzyme used, ST3Gal IV or ST6Gal I, respectively (Fig. 5). The human ST6Gal I clone (hST6 B1) yielded 11.5 ng 6′ sialyllactose considerably higher than the 3.9 ng of 6′ sialyllactose achieved with ST6Gal II. Chicken and stickleback ST6Gal I were also shown to yield 6′ sialyllactose. The efficiencies of these reactions were generally low, never achieving more

than a 6% conversion of lactose to sialyllactose, though better yields could have been achieved with more optimization.

### Enzyme-linked lectin assays

At the cellular level (Fig. 2) we demonstrated increased SNA-I labelling of CHO cells expressing ST6Gal I. At the molecular level and in vitro this can be demonstrated by the de novo sialylation of ASF. ASF in a microplate assay should show limited binding to SNA-I but after reaction with purified ST6Gal enzymes there should be increased labelling by SNA-I. There should also be reduced binding to the exposed Gal groups in ASF now hidden as a result of sialylation. Gal groups (Galα1-4GlcNAc and other Galα1-related oligosaccharides) typically bind the RCA I lectin. In two separate experiments we observed clear binding of SNA-I by ELLA after reacting ASF with the different purified SIATs (Fig. 6A). Chicken and rat ST6Gal I were as effective as the commercial (mouse) ST6Gal I for adding on SNA-I recognized sugars (probably Neu5Ac(α2-6) linked to Gal. The other ST6Gal I preparations were less effective and the ST6Gal II preparation showed no activity in this assay. As expected the ST3Gal IV preparations did not affect SNA-I binding as Neu5Ac(α2-3) linked to Gal is not an epitope for SNA-I. Reduced binding of RCA I was also observed for the strongest ST6Gal I preparations (Fig. 6B). The most effective enzyme in this assay was chicken ST6Gal I with approximately the same activity as the commercial mouse ST6Gal I.

## DISCUSSION

Sialyltransferases are very important enzymes that shape the terminal structures on glycoproteins and glycolipids and can have profound effects on how cellular surfaces interact with each other (*Varki, 2007*). The enzymes are therefore key players in determining how a cell recognizes a foreign bacterium or virus. SIATs also modify free oligosaccharides: new-born human milk contains high concentrations of oligosaccharides including the sialyllactoses that are synthesized from lactose by ST6Gal and ST3Gal enzymes. These human milk oligosaccharides offer protection against pathogens both directly in mopping up and excreting harmful bacteria and indirectly in promoting the colonization of health-promoting commensal bacteria (*Bode, 2012*).

We have developed a panel of stable CHO cell lines that express a variety of vertebrate SIATs. Most commercial enzymes from non-bacterial sources are of mammalian origin but we have chosen other species which have adapted to much more variable environments (Table S2). We have included homeotherms such as humans with a blood temperature of 36.5–37.5 °C, rats at 35.9–37.5 °C and chickens at 40.6–41.7 °C. We have included several fishes which are poikilotherms with body temperatures that will vary considerably based on environmental conditions. Three-spined sticklebacks may live in water temperatures of 0–29 °C though optimal temperatures are considered 10–20 °C (*Lefébure, Larsson & Byström, 2011*). Zebrafish are tropical fish, can survive temperatures of 6.2–41.7 °C but their optimal temperature is 18–28 °C (*Cortemeglia & Beitinger, 2005*). Fugu are temperate fish and their optimal temperature is 15–24 °C (*Nakajima, Nitta & Fujita, 2010*). Furthermore, zebrafish are freshwater fish where fugu

and stickleback adapt to freshwater and saltwater conditions. Quite possibly animals can survive at temperatures where their SIATs are barely functional but it would seem logical that the SIATs work efficiently at the optimal growth temperature of the animal. An isolated enzyme may however be more efficient at a different temperatures.

We have generated enzymes that are as comparable in structure to each other as possible by defining, for our purposes, the start of the catalytic domain (Table 1). It has been reported that N-terminal deletion between residues 90 and 100 in human ST6Gal I leads to a complete loss in sialyltransferase activity (Donadio et al., 2003) though this may depend on the expression system (Ribitsch et al., 2014). The full-length enzyme shows preference for some glycoproteins acceptors over others but this preference is lost when the catalytic domain is untethered from the membrane (Legaigneur et al., 2001). The untethered enzyme also shows increased activity as the stem is reduced to approximately residue 60 when it increases no further (Legaigneur et al., 2001). We therefore chose to retain some structure in the stem region. However, it is possible that in retaining more of the stem we have compromised protein stability as it is also known that the stem is susceptible to protease attack (Kitazume-Kawaguchi et al., 1999). We also included full-length human and zebrafish ST6Gal I sequences for comparative purposes. Both enzymes were secreted in the tissue culture media and the cell lines displayed SNA-I binding following lectin microscopy (Fig. S4). Our approach to expressing recombinant ST3Gal IV was similar. When ST3Gal I is deleted from the N-terminal from residue 57 to 77 enzymatic activity is lost (Vallejo-Ruiz et al., 2001).

Sequencing of our constructs identified several mutations, both missense and silent. These were most common in the human and zebrafish samples and rare in the other species, possibly as a consequence of long-term cultivation of the human (HepG2) and zebrafish sources in laboratories worldwide. We used proof-reading Taq polymerases throughout and saw no evidence that these mutations were introduced through amplification errors though this is still a possibility. Although reports of overexpression of human ST6GAL1 in cancer are common (Dall'Olio & Chiricolo, 2001; Kudo et al., 1998; Swindall et al., 2013), we are not aware of any record of these specific mutations in ST6GAL1. The NCBI SNP Report (Sherry et al., 2001) for human ST6GAL1 did however identify missense mutations directly adjacent to I295, I317, G327, and K395 which may suggest some accommodation for mutation in these particular regions. Although the functional consequences of these SNPs are not known, we would expect functional consequences for several of the mutations we identified as they were present in residues that are highly conserved across species. The 18 nucleotide insertion in ST3GAL4 (clone A1) is a recognized rare sequence so detection of this rare sequence variant in one of only two clones sequenced is of interest (Grahn & Larson, 2001). It would be surprising if there is not a functional consequence to this insertion as a potential N-glycan attachment site (Asn) is removed.

Multiple sialyltransferase activity assay approaches have been reported but we chose to use an indirect enzymatic approach for simplicity. Different acceptors (ASF, LacNAc, or Lac) were options but we focused on Lac (or LacNAc) because sialyllactose was our product of interest. Specific activity and $K_m$ measurements have been reported

for several SIATs (native and recombinant) using different approaches (*Datta & Paulson, 1995*; *Legaigneur et al., 2001*; *Ortiz-Soto & Seibel, 2016*; *Wu et al., 2011*) and our assay protocol was planned based on these values. Samples assayed were generally His-purified proteins from the media of stably transfected cells. Although ST6Gal activity in CHO cells is undetectable due to non-expression of the gene (*Lee, Roth & Paulson, 1989*; *Onitsuka et al., 2012*) ST3Gal activity is present and will interfere in this assay system when unpurified proteins are used. There is no specific acceptor available that can differentiate ST3Gal and ST6Gal activity in this assay. All ST6Gal I preparations of sequence-confirmed constructs gave detectable sialyltransferase activity. The two human ST3Gal IV clones and the human ST6Gal II clone showed marginal activity in this assay, though zebrafish ST3Gal IV gave appreciable activity and human ST3Gal IV (clone A1) did produce 3′ sialyllactose as determined by HILIC-HPLC. There is speculation that ST6Gal II prefers free oligosaccharides as acceptors over glycoproteins and glycolipids but lactose is not reported to be a substrate (*Krzewinski-Recchi et al., 2003*; *Takashima, Tsuji & Tsujimoto, 2002*). Production of 3′ sialyllactose and 6′ sialyllactose by incubation with the recombinant enzymes was confirmed by HILIC-HPLC although yields were low.

We also used three lectins to confirm the ability of the recombinant proteins to add Neu5Ac onto terminal Gal residues of ASF. Unfortunately biotin-labelled MAA binding was very low in this assay format and we could not confirm definite specific binding to Neu5Ac(α2-3)Gal structures on ASF (though we had no ST3Gal commercial positive control). Chicken and rat ST6Gal I enzymes yielded a glycoprotein that bound SNA-I much more strongly than ASF. As expected the same preparations showed reduced binding to RCA-I. Other SIAT preparations were not as effective but this could be a result of stability of the preparations or reduced affinity for ASF.

These stable cells may be used as a source of the purified sialyltransferase or they may be used to sialylate in vivo recombinant proteins. In vivo sialylation will depend on the compatibility of the growth conditions of the CHO cells expressing the recombinant protein of interest and the specific ST6Gal or ST3Gal stably expressed. For example, a recombinant protein that is known to be optimal in performance when produced at 30 °C may best be paired up with the stable cell line expressing a fish ST6Gal I whose activity is potentially favored by lower temperatures. In vitro reaction with the purified protein will be less restrictive as conditions are not limited by the growing CHO cell. Bacterial SIATs are very efficient at modifying oligosaccharides and, especially in so called "one pot" reactions, are the preferred means of making human milk oligosaccharides and many natural and unnatural sialosides. The promiscuity for modified donor Sias and a variety of acceptors, together with their high level expression, makes the bacterial enzymes ideal in this situation. However promiscuity is not an advantage when a precise sialylated structure is required: here the vertebrate enzymes have a distinct advantage.

Yields of the purified proteins were low but there is still scope for optimization. One major issue that limited production was the low adherence phenotype of these CHO cells in serum-free media. We have also not characterized the secretion profile of any

construct in terms of the optimal time to harvest the media. We have assumed that the FRT mechanism in the Flp-In-CHO cells ensures high stable expression so we have not select maximal producers. Since the native ST6GAL1 gene in CHO cells is not expressed it would be especially important to monitor the stability of these cell lines over time in culture (Onitsuka et al., 2012). The recombinant enzymes produced in this system have a direct effect on the host cells (as evidenced by lectin binding on growing CHO cells in Fig. 2) because the enzymes can potentially modify other cellular glycoproteins and glycolipids in the CHO cells. This could have a significant effect on CHO growth characteristics, however, it is not the reason for the semi-adherent growth we observed.

We have not yet fully characterized any of these enzymes in relation to substrate preference, $K_m$ for substrates, temperature, salinity, and pH optimums. Neu5Ac was the only Sia donor used in our study as it is the major donor in humans, however, Neu5Gc and KDN are important in other vertebrates. The expectation, based on their native environments, is that these proteins will show a range of biological and physiochemical characteristics which may be appropriate to specific opportunities in the Biopharma sector.

## ACKNOWLEDGEMENTS

We would like to thank John O'Connor, Glanbia Ingredients Ireland for providing the whey permeate.

### Funding

This research was supported by the Department of Agriculture, Food & Marine (grant nos. 13F477 and 15F747) and by Enterprise Ireland (grant no. CFR/2007/103). The funders had no role in study design, data collection and analysis, decision to publish, or preparation of the manuscript.

### Grant Disclosures

The following grant information was disclosed by the authors:
Department of Agriculture, Food & Marine: 13F477 and 15F747.
Enterprise Ireland: CFR/2007/103.

### Competing Interests

The authors declare that they have no competing interests.

### Author Contributions

- Benoit Houeix conceived and designed the experiments, performed the experiments, analyzed the data, prepared figures and/or tables, authored or reviewed drafts of the paper, approved the final draft.
- Michael T. Cairns conceived and designed the experiments, analyzed the data, prepared figures and/or tables, authored or reviewed drafts of the paper, approved the final draft.

## Data Availability

The research in this article did not generate any data or code. Any raw data (as in measurements) is presented in Figs. 2 and 5.

## Supplemental Information

Supplemental information for this article can be found online at http://dx.doi.org/10.7717/peerj.5788#supplemental-information.

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
