# Peer review of "Engineering of CHO cells for the production of vertebrate recombinant sialyltransferases"

_PeerJ, doi:10.7717/peerj.5788_

## Round 0.1 · original submission · Major Revisions

The manuscript was reviewed by two referees. Their detailed comments are in this email. I have examined the manuscript in light of these comments. As you can see, there are many significant concerns about the presentation and interpretation of the research work. I hope you will be able to address all of them by appropriately modifying the manuscript.

Reviewer 1 ·

Basic reporting

1. In figure 2 to 5, uniform name of samples should be used.
2. In two bars in figure 5(A), which one is Expt 1? Please add some explanations.
3. Statistical analysis (t-test) should be included in Figure 2 and 5.

Experimental design

The research question was defined as expression and characterization of a range of sialyltransferases. But I’m not convinced that expressed proteins were well characterized due to low protein yields, 1-33 micro-g, line 331 and Figure 3.
1. Expression system should be reconsidered to prepare recombinant sialyltransferases in better yield, and some experiments and results should be added to quantitatively characterize the expressed transferases, in terms of enzymatic parameters and protein stability.
2. In Figure 5, the authors performed enzyme-linked lectin assay to evaluate the enzymatic activities to glycoprotein. To ensure the activities, MALDI-TOF-MS or HPLC analysis of sugar chains from glycoproteins should be included.

Validity of the findings

I fail to understand the properties of expressed sialyltransferases through a manuscript.
1. Although recombinant proteins showed enzymatic activities, acceptors differed among experiments, LacNAc, lactose and asialofetuin (glycoproteins) for Figure 2, 4 and 5, respectively. I think it is better to describe the meaning of different acceptors.

2. Although expressed enzymes showed sialyltransferase activities (Figure 2 and 5), they seemed to take a few activities in Figure 4, and bands in WB analysis (Figure 3) were not clearly observed even in purified proteins. Because of these inconsistencies, I may fail to understand the properties of expressed sialyltransferase.

Additional comments

In this manuscript, the authors prepared a range of sialyltransferases and tried to characterize their enzymatic activity. They have a good attempt at assessing non-mammalian sialyltransferase. While this study represents a massive effort, prepared enzymes were not well characterized due to low protein yield (1-33 micro-g, line 331 and Figure 3). I think the author needs to reconsider the expression system to improve the yield, and add some experiments.
I hope that my comment is useful.

Reviewer 2 ·

Basic reporting

The title is not appropriate. What are these diverse environments mentioned in the title?
Quality of the presentation is poor and reported data are not rigorous and somewhat ambiguous. Controls (negative or positive) are missing in most of the proposed experiments.
A number of sections are missing which renders the manuscript difficult to review (no page number). A list of abbreviations and “Materials section” should be provided: pSecTag/FRT/V5-His-TOPO, What are FRT sites line 289 and FRT mechanism line 479? What is pOG44? Where does that come from? SNA-I, RCA-I MAA are from? Which MAA type?

The introduction section gives a very poor description of the context of the study and very limited field background. This section needs to be extensively revised and the research question clearly defined.

Line 56: Explain “SIATs are broadly split”
Lines 61-62: “Generally the stem is less structured and of varied length across SIAT families.” It varies across sialyltransferases themselves see ST6GalNAc I and ST6GalNAc II.
Lines 45-48: as reported recently in the mentioned reference Petit et al. 2018, sialyltransferase sequences are described across eukaryotes, in plants as well as in Bacteria. So “ST6Gal are the oldest” lines 72-73 is not true. In addition, the first ST3Gal enzymes (line75-76) appear much sooner than urochordates since they were already present in sponges (Petit et al. 2015 Mol Biol Evol 906-27).
Lines 69-71: Other crystal structures were obtained recently for the human ST8Sia III (Volkers et al. 2015 Nat Struct Mol Biol 627) and the human ST6GalNAc II (Moremen et al. 2018 Nat Chem Biol ). References should be updated. In this latter study, Moremen and collaborators describe an efficient expression system of glycosylation enzymes for their structural and functional studies. This should be mentioned
Lines 76-77: There are six ST3Gal paralogs in human tissues and eight ST3Gal paralogs in the chicken and zebrafish tissues (Teppa et al. 2016 Int J Mol Sci E1286) and for most of them we know nothing about kinetics and substrate specificities. It is important to describe exactly the sialyltransferases used in this study (sequences, accession numbers, species) and explain why these were chosen and not the others.
Line 95: Explain “specificity to substituted Sia donors”.

Experimental design

The experimental design is also extremely poor. The research question is not well defined. Furthermore, the material section is missing and the method section should clearly describe the research protocol and how measurements were made. There is no description of western blot experiments for instance.
Line 104: What are these selected protein sequences? The alignment need to be shown since it is somewhat difficult to imagine constructs “starting 121 amino acids N-terminal to a Cys that is totally conserved across all animal sialyltransferases (Cys-184 in human)”. In addition, this N-terminal region of vertebrate ST6Gal I is extremely variable (see Petit et al. 2010 JBC). What is the expected molecular weight of these recombinant proteins?
Line 112-114 Explain “Slight variations in ST6Gal1 (zebrafish) ending at LAL, ST6Gal1 (stickleback) ending at AGS and ST3Gal4 (zebrafish) with an additional QKKW are a consequence of an evolving cloning approach”
Line 119-125: It is necessary to describe which sialyltransferase sequence was chosen in which animal species and how molecular cloning was achieved from which tissue?
Line 131 “amplified genes” Line 256 “transfected gene” or cDNAs?
Line 152 It is not clear to me in which medium (with or without?) serum, the recombinant enzymes are produced.
Line 177 crude or purified sialyltransferases?
Line 195 Analysis of proteins on NuPAGE should be shown
Line 216 Explain the rationale using whey permeate as a sialyltransferase acceptor?

Validity of the findings

The results section resembles a methods section although it is obscure which recombinant enzymes are produced. A clear description of the sequences used in the study should be given with accession numbers. Are they soluble in the cell culture medium or still membrane bound (full length enzymes?)? Controls of their efficient production should be provided. It is not clear to me the purpose of using these mutant ST6Gal I proteins? Time course of enzyme activities should be shown and effect of temperatures should be shown. Data are not robust and controlled and conclusion are not well stated
Lines 258-261 “The full length gene has a transmembrane domain that attaches the catalytic domain…” ??...”the stem is cytoplasmic???? Delete this sentence
Line 265 what is human ST6Gal1 B1? Why was the sequence codon optimized?
Line 267 What are the cloned proteins? The multiple sequence alignment should be shown. What is Cys-184??
Line 276-281 It is difficult to understand what these human ST6Gal I sequences are?
Line 281 idem for zebrafish clones (of what?)? H111 K187 rat R105?
Line 287 ST3Gal4i1 what is it?
What about the other transfected cells lectin binding profile?
Explain strong binding of MAA (and not moderate) in figure 1.

Line 305: why asialofetuin? What is the amount used? This is not mentioned in the material and methods? These data are not shown, why?
In figure 2 why incubation reactions are only 30 min? Where is the Wild type human ST6Gal I control?
Line 313 explain the results may be influenced by the choice of acceptor?
Line 313-321: range of protein concentration?? Difficult to understand what is used as an enzyme source? The authors should provide a WB. Another problem is the indirect sialyltransferase assay based on phosphate detection. Controls should be provided since these assays produce high background signals.
Line 336 and 315 exactly the same activities obtained after enzyme purification?
Line 341-346. The samples need to be run on a single gel (figure 3). PNGase F digestion should be shown for all recombinant proteins. Sizes of recombinant enzymes should be equivalent since catalytic domain is almost the same.
Line 363 What are these optimizations?
What about the ST3Gal IV activities?
Line 365-369: Already mentioned.
Figure 5 the WT human sialyltransferase activity should be shown. What are Expt1 and Expt2? Why so much differences between the 2 experiments? What is ST3A1 and ST3A2 and ST3H6
What is the effect of temperature variations on enzyme activities? The human ST6Gal I activity at 37°C should be shown. The activities mentioned in the text are obtained in which conditions? Figure 4 should show all the data obtained for all enzymes at different temperature.
Line 477: low adherence of transfected CHO cells is not mentioned before. Data should be presented. What is the importance of this observation since the recombinant enzyme is produced in the cell culture medium?

Unless otherwise stated, proteins should be noted ST6Gal I and human gene ST6GAL1 and rat, mouse zebrafish genes st6gal1.

---

## Round 0.2 · accepted · Accept

Unfortunately, both referees who had examined the original manuscript were unavailable to assess the revision. I have therefore carefully gone through the referees' comments, the authors' rebuttal, and the changes made to the manuscript. All the points that were raised by the reviewers were important, and most of them, in my opinion, have been satisfactorily addressed in the revised manuscript and/or rebuttal. Two major concerns, which required experiment work, were not addressed in the revised manuscript – improving protein yields, and MS/HPLC for finer evaluation of enzymatic activities. As stated by the authors in their rebuttal, they are unable to perform these experiments because of lack of resources. While the results of these experiments would have strengthened the observations and conclusions that are presented in the manuscript, they are not absolutely essential for the manuscript to be publishable in PeerJ.

#